# Establishment and Application of CRISPR–Cas12a-Based Recombinase Polymerase Amplification and a Lateral Flow Dipstick and Fluorescence for the Detection and Distinction of Deformed Wing Virus Types A and B

**DOI:** 10.3390/v15102041

**Published:** 2023-10-01

**Authors:** Yuting Xiao, Dongliang Fei, Ming Li, Yueyu Ma, Mingxiao Ma

**Affiliations:** 1College of Animal Husbandry and Veterinary Medicine, Jinzhou Medical University, Jinzhou 121000, China; lnjzxyt521@163.com (Y.X.); liming-1979@163.com (M.L.); 2Experimental Animal Center of Jinzhou Medical University, Jinzhou 121000, China; dlfei7712@126.com (D.F.); nysygmgm@163.com (Y.M.)

**Keywords:** deformed wing virus, recombinase polymerase amplification assay, CRISPR–Cas12a, lateral flow dipstick, detection

## Abstract

Deformed wing virus (DWV) is one of the important pathogens of the honey bee (*Apis mellifera)*, which consists of three master variants: types A, B, and C. Among them, DWV types A (DWV-A) and B (DWV-B) are the most prevalent variants in honey bee colonies and have been linked to colony decline. DWV-A and DWV-B have different virulence, but it is difficult to distinguish them via traditional methods. In this study, we established a visual detection assay for DWV-A and DWV-B using recombinase polymerase amplification (RPA) and a lateral flow dipstick (LFD) coupled with the clustered regularly interspaced short palindromic repeats (CRISPR)–CRISPR-associated protein (Cas) 12a fluorescence system (RPA–CRISPR–Cas12a–LFD). The limit of detection of this system was ~6.5 × 10^0^ and 6.2 × 10^1^ copies/μL for DWV-A and DWV-B, respectively. The assays were specific and non-cross-reactive against other bee viruses, and the results could be visualized within 1 h. The assays were validated by extracting cDNA from 36 clinical samples of bees that were suspected to be infected with DWV. The findings were consistent with those of traditional reverse transcription–quantitative polymerase chain reaction, and the RPA–CRISPR–Cas12a assay showed the specific, sensitive, simple, and appropriate detection of DWV-A and DWV-B. This method can facilitate the visual and qualitative detection of DWV-A and DWV-B as well as the monitoring of different subtypes, thereby providing potentially better control and preventing current and future DWV outbreaks.

## 1. Introduction

The honey bee *Apis mellifera* is vital to the agricultural environment as a pollinator, with an estimated global economic value of >225 billion US dollars. However, studies have reported that honey bee colonies are severely threatened by deformed wing virus (DWV), especially in the presence of the external parasite *Varroa destructor*, which is one of the most well-characterized, globally distributed bee-infecting viruses [1,2,3]. DWV is a major pathogen that is detected in all developmental stages of honey bees and causes colony collapse disorder [3].

DWV is a member of the picorna-like insect virus family *Iflaviridae*, consisting of a 30 nm icosahedral particle with a single, positive-strand RNA genome [4]. Honey bees infected with DWV during development may exhibit wing deformities, whereas those infected with DWV in adulthood may have asymptomatic or symptomatic infections, characterized by a shortened abdomen, cuticle discoloration, and a reduced lifespan [5,6,7]. Three master variants of DWV have been identified: types A, B, and C [8,9,10]. The International Committee on Taxonomy of Viruses has further categorized DWV type A into two variants: DWV and Kakugo virus [11,12]. DWV-A and DWV-B infections are prevalent worldwide and have different virulence, with type B being the most virulent [13]. As DWV-C is the most recently described variant, its effects have not yet been elucidated [10]. DWV is mainly diagnosed based on reverse transcription–polymerase chain reaction (RT–PCR) technologies [8], such as RT–PCR, RT–quantitative PCR (qPCR), and SYBR Green qPCR [1,14]. However, owing to complex detection procedures, costly test equipment, and the requirement for qualified technical personnel, the clinical application of these methods is limited. As a novel isothermal nucleic acid amplification technology, recombinase polymerase amplification (RPA) has been demonstrated to be rapid, specific, sensitive, and cost-effective for detecting pathogens. The entire reaction can be accomplished at a constant temperature (optimally at 37°C–42°C) within a short time (20 min), making RPA a promising on-site detection method [15]. RPA has been employed for detecting pathogens such as bacteria, fungi, parasites, and viruses [16,17,18].

Clustered regularly interspaced short palindromic repeats (CRISPR)-associated protein (CRISPR–Cas) was developed for gene editing and detecting nucleic acids and has been used to diagnose infections [19,20]. CRISPR–Cas12a-based diagnosis involves a mixture of Cas12a and guide RNA (crRNA) with protospacer adjacent motif (PAM), which specifically binds target dsDNA to activate Cas12a for target DNA cleavage. Collateral cleavage is activated, resulting in the cleavage of the fluorescent reporter from the quencher and producing a detectable fluorescent signal [21]. Compared with RPA or PCR alone, a combination of RPA and CRISPR makes the highly sensitive Cas12a detection system (RPA–CRISPR–Cas12a) more specific, prevents the loss of trace nucleic acids [22], compensates for the disadvantages of RPA false positives, amplifies the cleavage signal of CRISPR, and reduces dye contamination generated during staining [23]. The RPA–CRISPR–Cas12a assay has been extensively used for detecting pathogens, such as plant RNA viruses, *Ehrlichia canis*, and *Anaplasma platys* [24,25]. The results of the RPA–CRISPR–Cas12a assay can be visually detected using a lateral flow dipstick (LFD) and directly interpreted by the naked eye within a few minutes [26]. LFDs are based on lateral flow technology using gold nanoparticles and can promptly detect amplification products, making results easy to observe [27]. To the best of our knowledge, no studies have reported the use of RPA and LFD coupled with the CRISPR–Cas12a fluorescence assay for detecting DWV-A and DWV-B.

To establish a novel method for detecting DWV, we first designed specific primers to amplify the target regions of the RdRp gene of DWV. Further, we synthesized CRISPR RNAs (crRNAs) targeting DWV-A and DWV-B. Then, we combined the RPA–CRISPR–Cas12a assay with LFD to detect and distinguish DWV-A and DWV-B. The sensitivity and specificity of our detection method were evaluated and its clinical performance was validated using field samples.

## 2. Materials and Methods

### 2.1. Viruses and Clinical Samples

DWV-A virus was preserved in our laboratory. DWV-B cDNA was donated by Professor Hou Chunsheng from China Bast Fiber Crop Research Institute. Israeli acute paralysis virus (IAPV), acute bee paralysis virus (ABPV), Black Queen cell virus (BQCV), and Chinese sacbrood virus (CSBV) cDNA were donated by Professor Magali Chabert-Ribière from the French National Food Safety Authority. Healthy bees were collected from Benxi, Liaoning Province. RT–PCR was used to confirm infection in bees [28]. In total, 36 clinical samples of bees suspected of DWV infection were randomly collected: 3-, 5-, 7- and 9-day-old bee larvae, Varroa-exposed and -nonexposed pupae, and symptomatic or asymptomatic worker bees and newly emerged bees from six different regions of Liaoning Province and Jilin Province. Details of the clinical samples are shown in Appendix A.

### 2.2. Construction of Standard Recombinant Plasmids

The nucleotide sequences of 14 DWV-A and DWV-B (DWV-A gene IDs: AJ489744, MZ821832, MH069503, MG831201, MN538208, MN746311, MT415949, MW222481, and AY292384; DWV-B gene IDs: JX878305, MT747986, KX783225, NC_006494, and AY251269) variants were aligned via Mega 6.06 to help design variant-specific and conserved fragments for plasmid assembly (Appendix A) [29]. AJ489744 (9146–9498 bp) and MT747986 (9153–9506 bp) were selected as conserved target gene sequences of the RdRp region in DWV-A and DWV-B, respectively. The conserved target gene sequences of DWV-A and DWV-B were synthesized and cloned into pUC57 vectors (Appendix A) as recombinant plasmids at Sangon Biotech (Shanghai, China) [30].

### 2.3. Design and Preparation of RPA Primers, crRNAs, and ssDNA Reporters

Three pairs of specific primers were designed for DWV-A and DWV-B, based on the target gene sequences of DWV-A and DWV-B and the principles of RPA primer design (Appendix A). crRNAs were designed to detect the gene sequences of DWV-A and DWV-B adjacent to the 5′-TTN-3′ and 5′-TTTN-3′ PAM site (where N refers to A/G/C), respectively (Appendix A). To prepare crRNA, synthetic crDNA, containing the T7 promoter, was used as a template, and crRNA was synthesized in vitro at 37 °C for 16 h using HiScribe T7 Quick High Yield RNA Synthesis Kit (NEB, MA, USA). Then, the transcription product was treated with DNase I at 37 °C for 1 h to degrade the dsDNA template. The crRNA was purified using the Monarch RNA Cleanup Kit (NEB, MA, USA). The concentration of the purified crRNA product was measured using a Qubit miRNA assay kit and Qubit 4 Fluorometer (Thermo Scientific, MA, USA). The crRNA product was stored at −80°C until use. To detect the need, we designed two types of reporters: an ssDNA reporter (5′FAM-TTATT-3′-BHQ) was used to detect CRISPR–Cas12a fluorescence, whereas a LFD ssDNA reporter (5′FAM-TTTTTTATTTTTT-3′-Biotin) was used to detect CRISPR–Cas12a–LFD. The primers and probes for target sequences were synthesized by Shanghai Sangon Co., Ltd. (Shanghai, China). The details of primers, crRNAs, and ssDNA reporters are presented in Table 1.

### 2.4. Establishment of the RPA Assay

RPA was performed in 50-µL reaction mixtures using TwistAmp Basic Kit (TwistDx, Maidenhead, United Kingdom). The composition of the stock reaction mix was as follows (working concentrations are indicated in parentheses): 29.5 µL of rehydration buffer, 1 µL of plasmid template, 2.4 µL of forward primer (10 µM), 2.4 µL of reverse primer (10 µM), 12.2 µL of dH_2_O, and 2.5 µL of magnesium acetate (280 mM). All reagents, except template and magnesium acetate, were prepared as a master mix, which was added into a 0.2 mL tube containing a dried enzyme pellet. Next, 1 µL of the template was gently pipetted into the reaction tubes; the internal and blank control templates were treated simultaneously. Finally, 2.5 µL of 280 mM magnesium acetate was added to each tube, and the tubes were covered tightly for a short spin to initiate the reaction. For amplification, the reaction mixture was incubated at 39 °C for 20 min. Subsequently, 20 μL of the amplified product was added to 40 μL of phenol solution, and the mixture was subjected to vortex centrifugation. The supernatant was run on 1% agarose gel and visualized under a UV transilluminator.

### 2.5. RPA–CRISPR–Cas12a Fluorescence-Based Detection Assay (FBDA)

The 20 μL CRISPR–Cas12a reaction mixture contained 1.5 µL of crRNA (10 µM), 0.5 µL of LbCas12a nuclease (10 µM) (TOLOBIO, Shanghai, China), 1 µL of ssDNA reporter (10 µM), 2 µL of HOLMES Buffer 1 (10×), 2 μL of RPA product, and 13 µL of nuclease-free water. The reaction mixture was incubated at 37.5 °C for 20 min, and the fluorescence (FAM) signal was detected every 20 s via QuantStudio1 (ABI, MA, United States). At the end of the assay, the reaction tube was examined under an LED transilluminator to visualize the fluorescence signal with the naked eye.

### 2.6. RPA–CRISPR–Cas12a Assay Combined with an LFD Assay

An LFD assay was established to improve the RPA–CRISPR–Cas12a assay, thereby making it easier to use the assay and interpret the results. The FAM–biotin ssDNA reporter and LFD were designed to capture labeled nucleic acids. The 20 μL RPA–CRISPR–Cas12a–LFD reaction mixture, containing 0.5 μL of LbCas12a nuclease (10 µM), 2 µL of HOLMES Buffer 1 (10×), 1 µL of ssDNA reporter (Sangon, Shanghai, China), 1.5 μL of crRNA A1 (10 µM), 13 μL of nuclease-free water, and 2 μL of RPA products, was mixed thoroughly and incubated at 37.5 °C for 20 min. Then, 10 μL of nuclease-free water was added to the reaction mixture and mixed thoroughly. The LFD was placed in the mixture for 3–5 min. If both the test and control lines were displayed, the result was considered positive. If only the control line was displayed, the result was considered negative. If the control line was not visible, the result was considered invalid, and the test was repeated using a new strip.

### 2.7. Validation with Clinical Samples

To evaluate the performance of RPA–CRISPR–Cas12a–LFDA on clinical samples, 36 samples were tested for DWV-A and DWV-B at different stages of bee development. Genomic RNA from clinical samples was extracted using the TIANamp Virus DNA/RNA Kit (Tiangen, Beijing, China) according to the manufacturer’s instructions. RNA was reverse-transcribed into cDNA using a first-strand cDNA synthesis kit (TransGen, Beijing, China) and then stored at −20 °C until use. cDNA extracted from the samples were analyzed using RPA–CRISPR–Cas12a–LFD assay and RT–qPCR. The method described by Bradford et al. was used to detect clinical samples via RT–qPCR [1]. The clinical samples were typed and tested, as described previously. The results were analyzed and compared with those of RT–qPCR. The primers used in this experiment are listed in Table 1.

### 2.8. Statistical Analysis

The agreement accessing between the two methods was calculated via Cohen’s “kappa” (κ) analysis, with a κ-value of ≥0.750 and *p*-value of <0.0005 denoting good agreement.

## 3. Results

### 3.1. Screening of RPA Primers and crRNA

According to the manufacturer’s instructions, primers with the brightest and most specific bands on the agarose gel were selected for subsequent experiments. Accordingly, the first set of primers (DWV-RPA-F1 and DWV-RPA-R) was selected for the experiments (Figure 1A). After isothermally amplifying the positive plasmid using RPA, the RPA–CRISPR–Cas12a fluorescence assay was used to screen the crRNAs. The amplification efficiency of the probes was evaluated based on a fluorescence amplification curve (Figure 1B,C). Finally, the crRNAs A1 and B1 were selected for the experiments.

### 3.2. Specificity and Sensitivity of the RPA–CRISPR–Cas12a-FBDA

The RPA reactions were performed using the cDNA of DWV-A, DWV-B, IAPV, ABPV, BQCV, and CSBV as templates. The RPA products were combined with CRISPR–Cas12a for the fluorescence assay. The results showed that only DWV-A or DWV-B was detected using the RPA–CRISPR–Cas12a–FBDA (Figure 2A,B).

To evaluate the possibility of cross-reactivity between the crRNAs designed for DWV-A and DWV-B, we conducted cross-reaction detection. The results demonstrated that crRNA A1 could only interact with DWV-A, whereas crRNA B1 could only interact with DWV-B. The primers of RPA and the crRNAs of CRISPR–Cas12a showed extremely high specificity without any cross-reactivity (Figure 3).

This finding demonstrates the specificity of RPA–CRISPR–Cas12a–FBDA for detecting DWV-A and DWV-B. To evaluate sensitivity, DNA plasmids of pUC57-DWV-A and pUC57-DWV-B were subjected to 10-fold serial dilutions to achieve concentrations ranging from 6.5 × 10^6^ to 6.5 × 10^0^ copies/μL and 6.2 × 10^6^ to 6.2 × 10^0^ copies/μL across seven gradients, respectively. The limit of detection (LOD) was 6.5 × 10^0^ and 6.2 × 10^1^ copies/μL, respectively (Figure 4A–D).

### 3.3. RPA–CRISPR–Cas12a–LFD Assay

To improve the RPA–CRISPR–Cas12a assay and make it more visual and easier to read for general use, we established the RPA–CRISPR–Cas12a–LFD assay. The LOD of this assay was 6.5 × 10^0^ copies/μL for DWV-A and 6.2 × 10^1^ copies/μL for DWV-B (Figure 4E,F). The results showed that the LFD assay could reliably distinguish known positive and negative samples.

### 3.4. Comparison of the Performance of the Assay and RT–qPCR in Clinical Samples

We evaluated the utility of the RPA–CRISPR–Cas12a–LFD assay for the rapid detection of DWV-A and DWV-B. We examined 36 samples of bees in different developmental stages (26 positive and 10 negative samples, as confirmed via traditional real-time PCR) and compared the results with those of RT–qPCR. RT–qPCR and the RPA–CRISPR–Cas12a–LFD assay (Table 2, Figure 5) were performed thrice for all samples.

The consistency in the results of the RPA–Cas12a–LFD assay and RT–qPCR for detecting DWV was 88.88% (32/36). Cohen’s “kappa” (κ) analysis revealed an extremely good agreement between the two methods, with a κ-value of ≥0.750 (κ = 0.753) and *p*-value of <0.0005 (*p* = 0.000) (Table 3).

## 4. Discussion

In recent years, DWV has become the most prevalent virus in honey bees, with a minimum average of 55% of colonies/apiaries infected across 32 countries [31], and is considered a major cause of colony decline [32]. DWV is a viral complex comprising at least three distinct genotypes or master variants—types A, B, and C—with types A and B being the most widespread variants [10]. The transmission efficiency of *V. destructor*, the main carrier and transmission medium of DWV, is different for each variant. A previous study showed that DWV-B, not DWV-A, infects mites and is more toxic to both individual bees and colonies than DWV-A [33]. Highly virulent and infectious DWV-B poses a major potential threat to wild bees and other nonvenomous bee colonies [34]. Mixed infections with DWV-A and DWV-B have greatly exacerbated bee mortality. Moreover, these variants are often mixed with other bee viruses such as BQCV, which is difficult to distinguish from DWV via single clinical diagnosis and serological tests. Therefore, the development of a rapid test for better visualization is key for distinguishing these two subtypes and performing epidemiological investigations of DWV. Therefore, the development of a qualitative visual detection method is key for helping bee farmers distinguish between these two subtypes and perform epidemiological investigations of DWV.

As a small RNA virus, DWV is more susceptible to mutations. Because DWV-A and DWV-B share high homology, distinguishing DWV-A subtypes from DWV-B subtypes is challenging [1,35,36]. We compared the nucleotide sequences of DWV-A and DWV-B isolates and revealed that these reference strains had a few relatively conserved gene regions, which were suitable for RPA [37]. We designed three primer pairs for RPA reaction based on these conserved regions. The results of agarose gel electrophoresis revealed that the first primer pair showed strong specificity and the brightest bands. Accordingly, we used this primer pair for subsequent experiments. In addition, multiple PAM sequences capable of undergoing CRISPR–Cas12a reactions were observed in this target segment. As the PAM sequences of DWV-A and DWV-B differed, we designed crRNA probes for DWV-A and DWV-B. Thus, we can distinguish DWV-A subtypes from DWV-B subtypes via CRISPR–Cas12a with high specificity. We amplified the target gene via RPA and used Cas12a to facilitate the binding of crRNA to target DNA, thereby forming a ternary complex, which activates transcleavage activity against nonspecific ssDNA [38]. Combining the advantages of the two methods enables trace detection. Therefore, the combination of RPA and CRISPR makes the highly sensitive Cas12a detection method more specific, prevents the loss of trace nucleic acids, compensates for the limitations of RPA false positives, and amplifies the lysis signal of CRISPR [23]. For 36 clinical samples of bees in different developmental stages, the consistency in the results of the RPA–CRISPR–Cas12a–LFD assay and RT–qPCR for detecting DWV-A and DWV-B was 88.88% (32/36). However, in bee larvae suspected of DWV infection, the detection rate of the LFD assay only reached 75%, probably because the viral copy number in bee larvae is low, resulting in the lower sensitivity of the RPA–CRISPR–Cas12a–LFD assay than that of RT–qPCR. The detection limits of RT–qPCR for DWV-A and DWV-B were reportedly 2.5 × 10^1^ copies/μL, whereas the detection limits of the newly developed assay were 6.5 × 10^0^ and 6.2 × 10^1^ copies/μL, respectively [1]. As expected, the sensitivity of the two methods was similar. A previous study reported that honey bee colony decline is frequently associated with increasing DWV loads [39]. DWV suppresses the bee’s immune system when the viral copy number reaches a certain threshold, promoting replication [40,41]. Although this method based on copy number allows for the rapid detection of viruses, demonstrates high adaptability, and has been verified in larger clinical samples, it cannot accurately quantify viruses. There are some disadvantages in assessing the copy number of viruses infected by bee colonies, and further modifications are needed for this method. Global DWV has evolved into four major lineages: DWV-A, DWV-B, DWV-C, and several recombinant variants. In China, a widely circulating DWV is DWV-A, and occasionally DWV-B; however, DWV-C and DWV-D have never appeared [42]. Based on this, we mainly focus on the visual, qualitative, and rapid detection of DWV-A and DWV-B to facilitate clinical application in China. In this study, RNA extraction and reverse transcription were performed in the laboratory; however, in the future, we aim to use a viral DNA/RNA rapid extraction kit for complete RNA extraction at the sampling site and perform follow-up reactions, thereby achieving actual rapid clinical detection. In this study, detection results in most clinical samples were identified as DWV-A, whereas all DWV-B variants were detected from the same bee colony in a particular region. Our test results are consistent with those reported in the literature. The monitoring of DWV-A and DWV-B may be a key factor for apiary management. This method may facilitate the development of point-of-care testing products in the future, especially for rural apiaries with limited resources, because the detection process is easy, practical, portable, and visual and requires fewer materials and specialized instruments.

In summary, using the combination of RPA and LFD in our established CRISPR–Cas12a fluorescence detection system, DWV-A and DWV-B subtypes can be distinguished and categorized to achieve rapid visual detection. The method is simple to operate, does not require expensive equipment, provides more intuitive reaction results, has strong specificity and high sensitivity, and can be performed at a constant temperature using a simple water bath or heating block or even at body temperature. This method is of great significance for the prevention, control, and eradication of DWV with strong virulence and coinfection and provides an early warning for the continuous spread of DWV to bee colonies. It can facilitate the healthy development of apiculture in the future. Therefore, it is expected to become an efficient and robust visual detection platform to accelerate DWV-A and DWV-B detection and develop containment strategies.

## Figures and Tables

**Figure 1 viruses-15-02041-f001:**
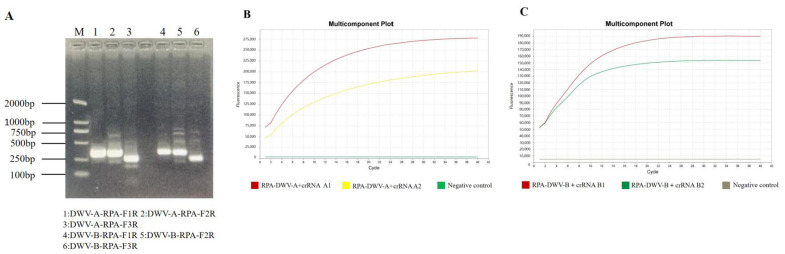
An RPA–CRISPR/Cas12a platform for the detection of DWV-A and DWV-B. (**A**) Screening of the optimal RPA primers for DWV-A and DWV-B. M: DNA marker DL2000; 1: DWV-A-RPA-F1R; 2: DWV-A-RPA-F2R; 3: DWV-A-RPA-F3R; 4: DWV-B-RPA-F1R; 5: DWV-B-RPA-F2R; and 6: DWV-B-RPA-F3R. (**B**) Screening of the optimal crRNA probe for DWV-A plasmid detection based on the RPA–CRISPR–Cas12a fluorescence amplification curve. (**C**) Screening of the optimal crRNA probe for DWV-B plasmid detection using the RPA–CRISPR–Cas12a fluorescence amplification curve. Negative control represents nontemplate control.

**Figure 2 viruses-15-02041-f002:**
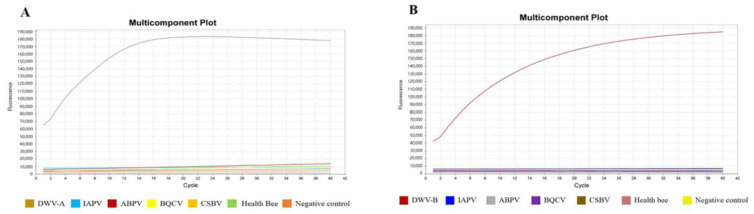
Specificity of the RPA–CRISPR/Cas12a-FBDA for the detection of DWV-A and DWV-B (**A**,**B**). Specific fluorescence signals were obtained for DWV-A and DWV-B, whereas no signals were obtained for other bee viruses or the control.

**Figure 3 viruses-15-02041-f003:**
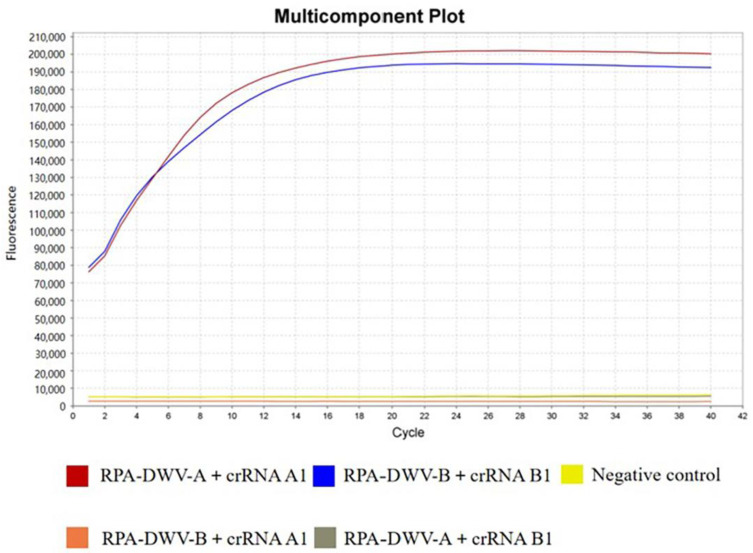
Cross-reactivity between the primers of RPA and the cRNAs of CRISPR–Cas12a designed for DWV-A and DWV-B.

**Figure 4 viruses-15-02041-f004:**
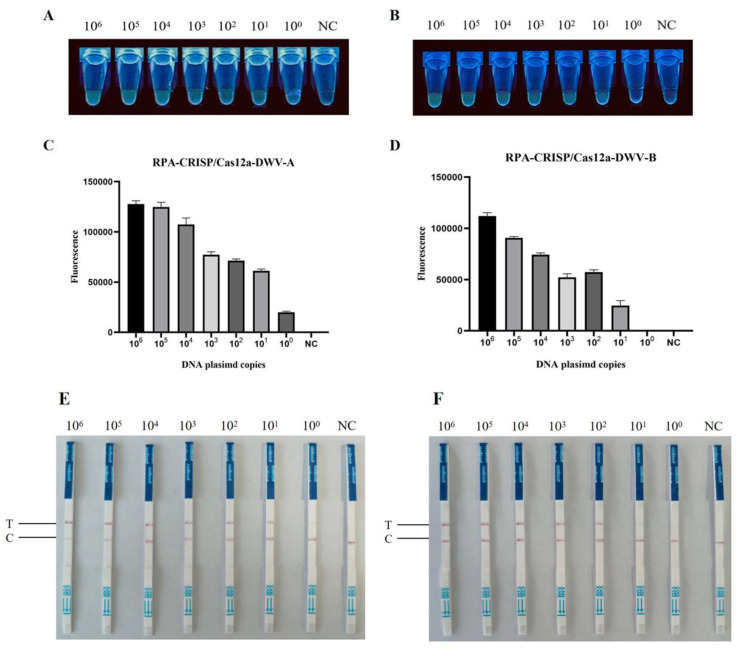
Sensitivity of the RPA–CRISPR–Cas12a detection assay. (**A**) Sensitivity of RPA–CRISPR–Cas12a–FBDA for the detection of DWV-A and (**B**) DWV-B. Serial 10-fold dilutions of DWV-A and DWV-B DNA plasmids (10^6^–10^0^ copies/μL) were detected. Detection using a UV transilluminator allowed examination with the naked eye. Comparison of fluorescence values generated via RPA–CRISPR–Cas12a-FBDA at each dilution (**C**): DWV-A, (**D**): DWV-B. NC represents nontemplate control. Sensitivity of the RPA–CRISPR–Cas12a–LFD assay for the detection of (**E**) DWV-A and (**F**) DWV-B.

**Figure 5 viruses-15-02041-f005:**
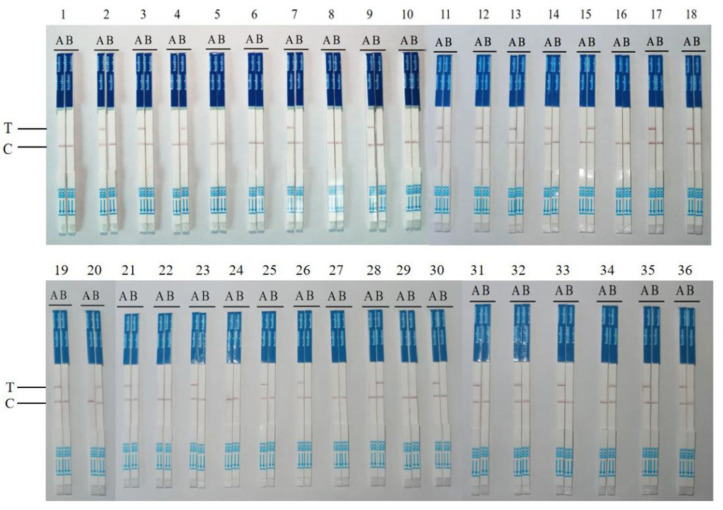
The RPA–CRISPR–Cas12a–LFD assay for the detection of clinical samples. The results of the cDNA samples detected from 36 clinical specimens in Liaoning, Jilin Province, and other regions. T represents the test band, whereas C represents the control band. The stick on the left side represents DWV-A, whereas that on the right side represents DWV-B.

**Table 1 viruses-15-02041-t001:** Primer, probe, and reporter sequences for RPA, crRNA, and CRISPR/Cas12a.

Name	Sequence (5′–3′)	Amplicon Size (bp)
DWV-RPA-R	GATTTRTCCTGATCCGTGAATTCCATCTTATAT	
DWV-RPA-F1	TAAAGACGAAATGAAGCGAGTAATGTGGAC	353
DWV-RPA-F2	AAATGAAGCGAGTAATGTGGACCATGGC	342
DWV-RPA-F3	ACCGAGTACCTTGTGGAATTCCATCAG	307
crRNA A1	ATCTCATGTTATCGCTAACATTCATGAATCTACAAGAGTAGAAATTcCCTATAGTGAGTCGTATTAatttc	
crRNA A2	ATCCAAAGGCAAATCAGTAATACCATCTACAAGAGTAGAAATTcCCTATAGTGAGTCGTATTAatttc	
crRNA B1	ATCCAAAGGCAAATCAGTAATACCATCTACAAGAGTAGAAATTcCCTATAGTGAGTCGTATTAatttc	
crRNA B2	ATCACCGTAACAAACTAGCACGACATGTCTAATCTACAAGAGTAGAAATTcCCTATAGTGAGTCGTATTAatttc	
DWV-FBD-ssDNA reporter	FAM-TTATT- BHQ	
DWV-LFD-ssDNA reporter	FAM-TTTTTTATTTTTT-Biotin	
DWV-A-RT-qPCR-F	GCGTGTTGCAACTCGCTTC	211
DWV-A-RT-qPCR-R	TGCCTGCACCGGATTCGATAAT	
DWV-B-RT-qPCR-F	GCAAGTTGGAGATAATTGTA	116
DWV-B-RT-qPCR-F	CGATACTTACATTCTTCAAGAT	

**Table 2 viruses-15-02041-t002:** Detection of DWV using the RPA–CRISPR–Cas12a–LFD assay in clinical samples, and a comparison of its performance with that of RT–qPCR. Ct: Average Ct for positive results. Type A: DWV-A; Type B: DWV-B; +: positive; −: negative. The numbers in parentheses indicate the number of positive results among the total number of replicates.

Sample ID	Sample Type	Assay Results			
		RT–qPCR		RPA–CRISPR–Cas12a–LFD	
		Type A or B	Ct	crRNA A1	crRNA B1
1	Bee larvae (3-day-old)	negative	negative	−	−
2	Bee larvae (3-day-old)	A	28.963	+(2/3)	−
3	Bee larvae (3-day-old)	A	31.631	+	−
4	Bee larvae (3-day-old)	B	34.123	−	+ (1/3)
5	Bee larvae (5-day-old)	negative	negative	−	−
6	Bee larvae (5-day-old)	negative	negative	−	−
7	Bee larvae (5-day-old)	A	26.347	+	−
8	Bee larvae (5-day-old)	negative	negative	−	−
9	Bee larvae (7-day-old)	A	27.933	+(1/3)	−
10	Bee larvae (7-day-old)	negative	negative	−	−
11	Bee larvae (7-day-old)	A	22.283	+	−
12	Bee larvae (7-day-old)	A	31.711	+(1/3)	−
13	Bee larvae (9-day-old)	A	20.028	+	−
14	Bee larvae (9-day-old)	A	23.672	+	−
15	Bee larvae (9-day-old)	B	29.431	−	+
16	Bee larvae (9-day-old)	negative	negative	−	−
17	Pupae (Varroa-exposed)	A	25.707	+	−
18	Pupae (Varroa-exposed)	A	17.373	+	−
19	Pupae (Varroa-exposed)	A	21.670	+	−
20	Pupae (Varroa-exposed)	B	23.824	−	+
21	Pupae (Varroa-nonexposed)	A	31.507	+	−
22	Pupae (Varroa-nonexposed)	negative	negative	−	−
23	Pupae (Varroa-nonexposed)	A	33.868	+	−
24	Pupae (Varroa-nonexposed)	negative	negative	−	−
25	Bee (newly emerged)	A	34.523	+	−
26	Bee (newly emerged)	A	21.639	+	−
27	Bee (newly emerged)	negative	negative	−	−
28	Bee (newly emerged)	B	30.706	−	+
29	Worker bee (asymptomatic)	negative	negative	−	−
30	Worker bee (asymptomatic)	A	28.416	+	−
31	Worker bee (asymptomatic)	A	30.933	+	−
32	Worker bee (asymptomatic)	A	28.215	+	−
33	Worker bee (symptomatic)	A	16.338	+	−
34	Worker bee (symptomatic)	B	24.949	−	+
35	Worker bee (symptomatic)	A	23.794	+	−
36	Worker bee (symptomatic)	A	20.347	+	−

**Table 3 viruses-15-02041-t003:** Statistical analysis of DWV detection in samples using the RPA–CRISPR–Cas12a–LFD assay and real-time RT–qPCR. κ = 75.3%, *p* = 0.000. The difference was statistically significant, as determined via the kappa test using SPSS software.

		RT–qPCR			CR
		Positive	Negative	Total	
RPA–CRISPR– Cas12a–LFD	Positive	22	0	22	88.88%
	Negative	4	10	14	
	Total	26	10	36	

## Data Availability

The original contributions presented in the study are included in the article/Appendix A; further inquiries can be directed to the corresponding authors.

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
