# Peer review of "Establishment and Application of CRISPR–Cas12a-Based Recombinase Polymerase Amplification and a Lateral Flow Dipstick and Fluorescence for the Detection and Distinction of Deformed Wing Virus Types A and B"

_viruses, 2023, doi:10.3390/v15102041_

Round 1
Reviewer 1 Report
This manuscript developed a rapid visual detection assay using recombinase polymerase amplification (RPA) and lateral flow dipstick (LFD) coupled with CRISPR–Cas12a fluorescence (RPA–CRISPR–Cas12a–LFD). The limit of detection for this system is comparable to qTR-PCR. The RPA–CRISPR–Cas12a assay is more specific, sensitive, rapid, simple, and can distinguish DWV-A from DWV-B. So, I suggest this paper can be published in the Viruses after minor revision.
minor concerns:
1.This manuscript should be polished by English speaker before publishing.
2. Check Figure S2, (C) and (D) or (E) and (F)?
3. Figure S1C,D should be S2?
4.Table 1, DWV-RPA-R should be listed once.
5. In Table 2, Bee larvea should be Bee larvae.
6. The primers for RT-qPCR should be listed in the MS.
Please edit the language properly.
Reviewer 2 Report
The detailed comments are listed in the attachment.

Minor editing of English language is required.
Reviewer 3 Report
In the submitted manuscript the authors present a RPA–CRISPR–Cas12a–LFD protocol for the detection of DWV. Based on the selected crRNA probes/primers, this method should be suitable to differentiate between DWV-A and DWV-B variants. The authors conclude that this method can facilitate point-of-care testing.
However, I have several reservations about this study, in particular to the material and methods and the results.
First of all, DWV is a highly mutating virus. DWV-A and DWV-B belong to a highly variable mutant cloud consisting of many variants including DWV-C and DWV-D. Beyond the sequenced and deposited variants, there are multiple minor variants circulating within every mutant cloud. Therefore, it cannot be expected that using two different sets of Primers or probes is sufficient to differentiate between the two variants DWV-A and DWV-B. It is not black or white only. There will always be intermediate variants. Because of this, it is not only expectable but also more likely that intermediate samples will be detected, having both, DWV-A and DWV-B. But according to the 36 field samples, there is no single sample showing both Virus variants. This issue needs to be clarified.
How does a “mixed infection” look like in the RPA–CRISPR–Cas12a–LFD? According to line 91, all 3-days-old larvae originate from one source (Kunadian). The results of the RPA–CRISPR–Cas12a–LFD show that there is DWV-A or DWV-B present in larvae which were collected serially from one colony. This is very unlikely. I would expect either DWV-A or DWV-B for all (four) larvae or both viruses together in one larvae. Or did the authors collect the four bee larvae (3-days-old) from different colonies? If so, it has to be given in more details in the manuscript.
Furthermore, the authors write, that the crRNAs were targeting DWV-A and DWV-B and cRNAs were designed using BLAST to detect the DWV-A and DWV-B gene sequences (line 108). When blasting crRNA A1, crRNA A2, crRNA B1 (table1) no DWV can be found in the BLAST databank. The only sequence which was positive for DWV was crRNA B2. But the authors selected crRNA A1 and B1 for the experiments. According to the sequences of crRNA A1 and crRNA B1, it is impossible to detect DWV at all. How did the PA–CRISPR–Cas12a–LFDA work when incorrect sequences where used?
Line 25: The authors claim that this method can facilitate point-of-care testing. But according to 2.7. the user has to do RNA extraction, RNA transcription and cDNA synthesis before RPA–CRISPR–Cas12a–LFD can start. This is far beyond point-of-care testing.
Line 86: What does it mean, that DWV-A was from the laboratory? Did the authors establish an infectious clone of DWV? Which bees were used for virus isolation? How did the authors differentiate between DWV-A and DWV-B?
Line 90: What are specific tests?
Line 90: What do the authors mean by clinical samples? This has to be described in more detail.
Line 91: It is impossible to use 3-year, 5-year, 7-year or 9-year-old larvae. The authors surely mean days instead of years.
Line 95: A reference is missing.
Line 104: References are needed. The material and methods of cloning-procedure are missing.
Line 116: What is the Cas12a assay? Please describe the methods in more detail.
Line 166: Hu et al. is not reference number 1 and the study is not listed in the references. Hu et al. is not listed in the reference list anyway. It is necessary to give the correct reference, otherwise it cannot be proved, if the RT-qPCR was suitable as reference method. Which primers were used in the RT-qPCR? The primers should be specific for the RdRP sequences as well.
Figure 1A: The figure legends are not comprehensible. Why is DWV-A-RPA-F1R used for 1, 2 and 3? What are the samples used for 1-6?
Table 2: According to the literature it should be expected that all symptomatic honey bees are positive for DWV-B only. But symptomatic worker bees 33-36 are positive for DWV-A and only positive for DWV-B in one sample. The authors have to discuss this issue. Furthermore, the authors have to reanalyze the (DWV-A) samples by using other primers or by re-sequencing to be sure that the samples are in actually DWV-A or DWV-B.
The results rather show that the RPA–CRISPR–Cas12a–LFD method is not specific for DWV-A or DWV-B.
Figure 5: The authors do not show 36 samples but they show only 24 samples.
The authors show one LFD per sample. How is it possible to differentiate between DWV-A and DWV-B by using only on LFD strip? Actually the authors have to show two sticks per sample.
Figure S1: This is not a Varroa destructor mite shown in figure S1B. The left pupae is non exposed and the right pupae is exposed to a mite.
Figure S2: The figure parts are named A, B, E and F. But it has to be A, B, C, D.
Round 2
Reviewer 3 Report
I rejected this manuscript because of several major issues and substantial errors in the material and methods. The other two reviewers recommended a minor revision only and therefore, it is not easy to recommend a rejection, I understand this. I realized that the authors changed some issues and revised the text but for me, I will reject this review again because the results do not match the current understanding of the biology behind this issue. Basically, there is no result showing a "mixed" infection. The virus DWV is a mutant cloud and therefore, I would expect test strips showing signals for both viruses. I already mentioned this but the authors did not argue convincingly.